# First Account of the Breeding Biology of Indian Blue Robin (*Larvivora brunnea*) in Southwest China

**DOI:** 10.3390/ani14010039

**Published:** 2023-12-21

**Authors:** Jun Nie, Shixiang Fan, Xu Luo

**Affiliations:** 1Faculty of Biodiversity and Conservation, Southwest Forestry University, Kunming 650224, China; swfuybnj@163.com (J.N.); shixiang_fan@163.com (S.F.); 2Key Laboratory for Forest Resources Conservation and Utilization in the Southwest Mountains of China, Ministry of Education, Southwest Forestry University, Kunming 650224, China

**Keywords:** *Larvivora brunnea*, breeding biology, life history characteristics, Laojun mountains, China

## Abstract

**Simple Summary:**

Breeding biology stands as the paramount component within avian life history. In this study, for the first time, we provided detailed information on the breeding biology of the Indian Blue Robin (*Larvivora brunnea*), a little-known forest understory bird, for which the breeding biology has been largely unknown. We conducted our fieldwork in northwestern Yunnan, China, in May of 2021. Both parent birds were observed to participate in the breeding process, but the female participated in incubation only. This study highlights how a small single-parent bird adapts its incubation behavior, as well as how parent birds coordinate their breeding to adjust to the cold and unpredictable environmental conditions in high-altitude regions.

**Abstract:**

Breeding biology lies at the core of life history research on birds, and it provides important information for avian conservation. We discovered one nest of the Indian Blue Robin (*Larvivora brunnea*) on 28 May 2021, at the Laojun mountains in Lijiang, northwestern Yunnan, China. Field observation was combined with the use of a GoPro camera for video shooting to quantitatively study the incubation and brooding behavior. We also conducted measurements of the eggs and nestlings on site and inspected the nesting materials used. A bowl-shaped nest with four eggs was located at 2830 m in the evergreen deciduous broad-leaved forest. All eggs were successfully incubated and two nestlings fledged on 22 June 2021, resulting in a total breeding success of 50%. Only the female bird incubated the eggs and brooded the nestlings. The incubation period was at least 13 days and the nestling period was 13 days. As incubation progressed, the incubation bout duration decreased. During the incubation period, the nesting time of the female bird shows a declining trend as incubation progresses. Both parents participated in feeding the nestling, and the frequency of parental supply increased with the maturity of the nestling.

## 1. Introduction

Understanding the life history traits of birds is important to gain insights into how they adapt to different environments [1]. Among all bird species worldwide [2], the life history traits of certain groups are disproportionally understudied [3]. The genus *Larvivora* encompasses a total of eight small foresting understory songbird species [4]. Probably due to their cryptic behavior in the forest understory, and the rarity of most species, data regarding the life history traits of species in this group, particularly those that breed in montane regions, are notably sparse. Indeed, detailed documentation of life history traits has been recorded for two species, including the Siberian Blue Robin (*L. cyane*) and the Japanese Robin (*L. akahige*) [5,6,7,8]. This information encompasses various life history traits, such as nesting sites, nest dimensions, clutch size, and egg morphology, as well as breeding behaviors in the incubation and nestling periods [5,6,7,8]. But our understanding of other species is far scanter.

The Indian Blue Robin (*L. brunnea*) is such an example. This species is classified under the avian order Passeriformes and the family Muscicapidae, with obvious sexual dimorphism. This indicates that there are distinct differences in physical morphology between males and females. This species ranges from eastern Afghanistan along the Himalayas to central China, overwinters in southern India and Sri Lanka, and inhabits intact forests with dense thickets at an altitude ranging from 1600 to 3300 m [8]. Despite its relatively large breeding region, scattered sightings have been documented in the Indian state of Gujarat [9], and more recent records in China were detected using infrared cameras in the Wujiao Nature Reserve in Sichuan Province [10]. Recently, a few records of the Indian Blue Robin have been documented at its stopover site during migration, including Yancheng City on the China’s eastern coast [11]. Thus, it is not so surprising that, according to Birds of the World [12], only the breeding season, nests, clutch size (3–4 eggs), and egg color (pale-blue color) have been described for this species thus far [12].

In the present study, we conducted detailed observations and made recordings of the nest, eggs, nestlings, fledglings, and parental behaviors of the Indian Blue Robin from a single nest discovered in the Laojun mountains of southwest China. From the video recordings, we extracted data on the incubation rhythm, feeding frequency, and nest cleaning behaviors. We also analyzed the incubation bouts and brooding bouts of any incubating bird. Finally, we compared our findings with other species in the *Larvivora* genus, specifically examining the variations in egg size among mainland breeding *Larvivora* species with differing elevational ranges, as well as the reproductive strategies of two island species, the Japanese Robin and Ryukyu Robin (*L. komadori*) [13]. We predict that: (1) throughout the entire breeding process, only the females engage in incubation; (2) there are no significant differences in parental feeding investment toward the offspring; (3) changes in egg size and clutch size in the *Larvivora* genus adhere to predictions of life history theory.

## 2. Materials and Methods

### 2.1. Study Site

We conducted this study in the Laojun mountains in southwest China. The Laojun mountains (26°2′48″–27°36′36″ N, 99°1′2″–99°54′36″ E) are major components of the Hengduan mountains, which form a vast area with a series of large mountains stretching from north to south at the east edge to the Qinghai–Tibet Plateau. The Laojun mountains divide the watershed of the Jinsha River (the upper portion of the Yangtze River) and the Lancang River (also known as the upper part of the Mekong River) [14]. The region is also an important area of “world biodiversity hotspots” (the mountains of Southwest China) [15].

The climate of the Laojun mountains is greatly influenced by the southwest monsoon from the Indian Ocean, the southeast monsoon from the Pacific Ocean, and the air mass from the Qinghai–Tibet Plateau. The average annual temperature is 12.75 °C and the annual average precipitation is 964.68 mm [16]. The highest peak stands at 4513 m, while the lowest elevation is approximately 1817 m. The vertical zonation of vegetation is remarkable, composed of evergreen broad-leaved forest, deciduous broad-leaved forest, coniferous forest, rhododendron shrubs, and alpine meadow toward the peak. This study was carried out in coniferous and broad-leaved mixed forests at an elevation of 2830 m [17].

### 2.2. Field Procedures

We found the nest of an Indian Blue Robin on 28 May 2021, during an avian survey on the existing trails in the Laojun mountains. When we found the nest, the egg laying had been completed, and the bird had started incubating the eggs. We noted the elevation at the nest using a Garmin GPS device (GPSMAP 62sc, Garmin Corporation, Taiwan, China). We then regularly checked the nest daily to confirm the breeding status (i.e., incubation, feeding, fledging, etc.) of this nest. Interestingly, we found a “bird blind” near the breeding nest, which refers to the practice of attracting birds by modifying microhabitats, providing water and supplementary food, and setting up concealed devices for bird photographers and birdwatchers. The presence of the bird blind caused the Indian Blue Robin parents to frequently bring worms to feed their nestlings.

In the incubation period, we carefully measured the eggs, when the incubating parent was absent from the nest to minimize the disturbance to the birds. We took the eggs out carefully and weighed each of them using an electronic balance (Wuxin Weighing Apparatus Co., Ltd., 20 g ± 0.001 g). We also used an electronic vernier caliper (Menet MNT-150T, precision of 0.01 mm) to measure the length and width of each egg. To ensure the reliability of the data, all operations were performed by the same person.

To monitor the incubation behaviors of the parent with minimum disturbance, we installed a GoPro camera (HERO5, GoPro, San Mateo, CA, USA) at a distance of 1–1.5 m away from the nest. We recorded videos at a rate of 30 frames per second. Our videotaping approach was limited by the camera’s battery life. On average, we obtained 271.7 ± 90.1 min (range: 119–439 min/d) of video footage per day. Moreover, when the parent bird was in the nest, the camera was not installed or changed. Instead, we made additional direct observations from a hidden location that was ~5 m away using binoculars. This approach yielded an average of 126.5 ± 77.4 min (range: 7–273 min/d) of observation per day. Equally, to minimize our interference, we have taken measures such as camouflage and concealing video equipment and observers.

We also measured the morphological traits of the nestlings every other day until they fledged. The measured morphological traits included body length, wing length, culmen length, tarsus length, claw length, and tail length. The measurements were performed consistently for every nestling. For example, the wing length (right-wing, flattened wind chord) was measured to the nearest 0.01 mm using a ruler cut off at the zero line, while culmen and tarsus length were measured to the nearest 0.01 mm using a sliding caliper. The culmen length was measured from the bill tip to the ceres, and the tarsometatarsus (tarsus) length was measured from the inner bend of the tibiotarsal articulation to the base of the toes [18]. The nestlings were measured according to body weight and several morphological measurements at 1, 3, 5, 7, 9, and 11 days. Again, to minimize the disturbance, we only measured the nestlings when the parent bird was absent from the nest. For the measurements of the morphological traits of the nestlings, we had assistance from another person who had received professional training.

After the breeding was completed, we brought the nest back to the laboratory to measure its size. Specifically, we took great care in handling the extraction and transportation process to prevent any deformation of the nest. We securely fixed the nest using rigid cardboard boxes and foam boards, ensuring no deformation occurred during transportation. We measured the nest height, nest depth, inter diameter, and outer diameter, using a sliding caliper measuring to the nearest 0.01 mm.

### 2.3. Data Analysis

By extracting video data and our direct observations, we recorded the duration of the parent birds’ incubation, feeding, and brooding behavior each day. We defined the incubation bout as the time interval between when the parent bird enters the nest and when it leaves during the incubation period. Similarly, we defined brooding bouts during the nestling period. We also noted that parent birds jointly feed their offspring. We defined the time interval between the previous feeding and the next feeding as the feeding interval, and the number of feedings per day by the parent birds as the feeding frequency.

We used a generalized linear regression model to assess how the incubation bout changed across the incubating day. In this model, incubation bout was the response variable, and recorded incubation was the predictor. Similarly, we used a model in which brooding bout was the response variable and age of the nestlings was the predictor, to demonstrate how brooding bouts change with the age of the nestlings. We analyzed our data using Employ GraphPad Prism v8 and SPSS 26. Values are presented as means (± SD).

## 3. Results

### 3.1. Nest and Eggs

The nest site was located in a ground cave on a slope (26°48′33.67″ N, 99°36′50.55″ E; 2830 m). The surrounding vegetation is mainly composed of dense evergreen deciduous broad-leaved forest with some bamboo stands, and the height of vegetation is 2–8 m (Figure 1a). The nest was bowl-shaped. It was covered with the dead leaves of weeds. The nest primarily comprised slender grass stalks, withered foliage, wool, and moss. Its outer nest diameter was 8.2 cm, while the inner cavity had a depth of 2.9 cm and a diameter of 4.0 cm. Four eggs were laid when the nest was found on 28 May 2021 and all four eggs survived through incubation. The eggs were oval, plain, pale blue, and spotless. The dimensions of the four eggs were 19.87 × 14.76 mm, 2.182 g; 20.35 × 15.19 mm, 2.365 g; 19.81 × 14.51 mm, 2.112 g; and 19.73 × 14.78 mm, 2.173 g (Figure 1d).

### 3.2. Incubation

Throughout the incubation period, we combined videotaping and field observation to monitor the parental behavior of the nestlings. In total, 2445 min of video and 1223 min of observation were recorded. On average, the incubating birds were monitored for 407.6 ± 100.7 min (range: 273–533 min/d) per day. Our results support prediction 1. We found that the duty of incubation was solely performed by the female bird. We recorded a total of 3011 min of incubation time by the female bird, with an average daily incubation duration of 301.1 ± 98.0 min (range: 176–444 min/d). In contrast, the male occasionally gave an alarm call to the incubating female nearby. The incubation lasted at least 13 days. The average incubation bout was 29.5 ± 21.7 min (range = 1–115, *n* = 102 bunts), which showed a downward trend over time (*R*^2^ = 0.0968; *p =* 0.0015) (Figure 2a).

### 3.3. Parental Care, Growth of Nestlings, and Breeding Success

During the feeding period, we used the same approach (videotaping and field observation) to monitor the parental care behaviors. In total, 1881 min of video and 674 min of observation were recorded. On average, the parental care behavior was monitored for 425.8 ± 114.4 min (range: 219–588 min/d) per day. We found that only the female bird brooded the nestlings and that its brooding time decreased as the nestling’s age increased (Figure 2b). A total of 363 feeding events were recorded with an average time interval of 11 ± 14.4 min (range = 0–138, *n* = 356). The feeding food was mainly insect larvae. With an increase in nestling age, the parental feeding frequency increased. During the entire process, it was noted that both parents jointly facilitated the provision of nourishment and the elimination of fecal sacs, with no marked gender difference between them (females = 198; males = 165) (supporting prediction 2). We found that the fecal sacs of the fledglings were generally ingested or transported by their caretakers (*n* = 41). The fledglings’ eyes were observed to remain closed until the seventh day. All four eggs incubated successfully on 9 June 2023; however, only two birds were successfully fledged on 22 June (Figure 1f). The other two nestlings disappeared for unknown reasons on 18 June.

### 3.4. Development of Nestlings

On the first day, the weights of the four nestlings were 2.366 ± 0.143 g (range =2.155–2.556, *n* = 4). The body weight continued to increase, and the tarsus length of the nestlings was as long as an adult male’s at 11 days of age (Table 1). At hatching, the nestlings had a pink complexion and a yellow beak with a few dark feathers on the head, back, and wings (Figure 1e). When the nestlings fledged, their tail feathers were still short and their feet were pink, leaving only a small yellow patch at the base of their beak (Figure 1f). The growth of ventral feathers was still in its infancy (about 75% coverage).

**Table 1 animals-14-00039-t001:** Morphological measurements of nestlings and adults of Indian Blue Robin.

Body Measurements	Body Mass/g	Total Length/mm	Culmen /mm	Wing /mm	Tail/mm	Tarsus/mm
1-day-old nestling ^a^	2.349	32.88	3.91	11.14	0	5.87
3-day-old nestling ^a^	5.173	42.62	4.66	14.98	0	10.53
5-day-old nestling ^a^	8.780	55.82	6.13	25.28	0	14.78
7-day-old nestling ^a^	13.462	59.65	6.75	31.74	2.45	22.74
9-day-old nestling ^a^	15.417	62.96	8.71	38.24	4.46	25.57
11-day-old nestling ^a^	16.125	66.33	9.06	54.22	8.98	28.83
Adult male ^b^	12–18	126–152	11–14	72–83	44–66	25–30
Adult female ^b^	14–14	125–125	12–12	73–73	51–51	25–25

^a^: This study; ^b^: [19].

## 4. Discussion

Our study provides the first detailed information on the breeding biology of the Indian Blue Robin, including its nests, eggs, nestlings, parental incubation, and brooding. It included information on the nesting site and nest material, the morphological details for fresh eggs, and the parental care behavior as well, which all in all filled in the non-existing breeding information for this species. The incubation period lasted for at least 13 days and the nestling period was 13 days. The eggs were oval, plain, pale blue, and spotless. The Indian Blue Robin exhibited a hatchability success rate of 100%, with a fledging rate of 50%. Our video failed to record their disappearance, which may have been caused by nest predators. When the nestlings matured, both male and female birds were seen feeding them, and the frequency of feedings increased.

The Indian Blue Robin is a sexually dimorphic bird species, with the males displaying vividly colorful plumage while the females exhibit a more subdued feathering (Figure 1b,c). Throughout the incubation period, our observations revealed that exclusive incubation duties were carried out by the female bird, while the male bird provided a restricted amount of sustenance to its mate. Soler and Moreno [20] assessed the life history traits of European passerine birds and revealed that there is no clear relationship between incubation attendance and plumage conspicuousness, irrespective of whether the nest type is open or concealed in a cavity. More recently, multiple studies have indicated that the bright coloration ornamentation of parents poses a significant risk to their offspring, as it reduces the nest’s ability to provide concealment and defense, thereby lowering the chances of successful breeding [21,22]. During the day, the female bird engaged in incubation activities only, a pattern that was found in other species breeding, such as the Fire-Tailed Sunbird (*Aethopyga ignicauda*) in the southwest mountainous region [23]. This may be because male birds entering the nests are more easily exposed, thus increasing the risk of predation [21]. Thus, the male birds may plausibly restore equilibrium in parental investment by feeding the female bird or employing a multitude of strategies, including nest construction and vigilant behavior during the breeding process.

Life history theory predicts that high-altitude birds lay fewer and larger eggs, while low-altitude birds lay more and smaller eggs [24,25]. By comparing our findings with those of six species in the *Larvivora* genus, we find the Siberian Blue Robin *(L. cyane*) [4,5,6] and Rufous-Tailed Robin (*L. sibilans*) [26,27] bred at low altitudes and laid more, albeit smaller, eggs, which is consistent with the theory predictions. Meanwhile, the Indian Blue Robin and Rufous-Headed Robin [19,27], breeding at high altitudes, laid eggs of intermediate size and quantity (Table 2). Thus, our results are not entirely in line with the previous predictions. It is noteworthy that the Japanese Robin [7] and the Ryukyu Robin [14] exhibit a unique reproductive strategy on low-altitude islands, characterized by fewer but larger eggs. Studies have indicated that island avian species typically exhibit lower fecundity, greater reproductive investment, and longer developmental periods [28]. As the population density expands, intra-specific competition on islands can intensify, thereby favoring a reduction in the number of eggs produced. The size of eggs often exhibits a negative correlation with their quantity, which is recognized as an energetic trade-off [29]. Island species are inclined toward prioritizing the quality of their offspring over quantity (or “K-selected”) [28,30], to enhance the survival rates of their progeny [31]. Recent reviews show that the influencing factors on bird life history are not only altitude but also include the inherent biological characteristics of species, phylogenetic relationships, and environmental interactions [32,33,34] (prediction 3 is not supported). It should be noted that our sample size for the study was very small, so further research is needed to confirm these patterns.

**Table 2 animals-14-00039-t002:** The comparison of breeding characteristics for the eight *Larvivora* species.

Species	Nesting Site		Nest		Egg				Incubation Period (d)	Nestling Period (d)	Breeding Month	References
	Elevation (m)	Habitat	Clutch Size	Nest Material	Fresh Mass (g)	Size (mm)	Color	Pattern				
Japanese Robin (*Larvivora akahige)*	1000–2500	Broadleaf evergreen forest, mixed conifer–broadleaf forest	4–5	Bryophyte and dead leaves	-	22.05 × 16.25	Pale greenish–blue	No speckles	12–15	12	5–7	[7]
Ryukyu Robin (*Larvivora komadori*)	1100–1600	Broadleaf evergreen forest, undergrowth	3–5	Bryophyte and bamboo leaves	-	22.5 × 17	Pinkish	No speckles	-	-	4–8	[13]
Okinawa Robin (*Larvivora namiyei*)	1100–1600	Broadleaf evergreen forest	-	-	-	-	-	-	-	-	-	[35]
Izu Robin (*Larvivora tanensis*)	-	Occurs in dense undergrowth of mature or old-growth damp, montane, deciduous, and evergreen temperate forest	-	-	-	-	-	-	-	-	-	[36]
Rufous-tailed Robin (*Larvivora sibilans*)	1200	Damp broadleaf evergreen and semi-eve rgreen bottomland forest with dense undergrowth	4–6	Bryophyte, grass stems, and pine needles	-	19–20 × 14–16	Pale blue, grayish blue	brown blotches	-	-	6–7	[19,26]
Rufous-headed Robin (*Larvivora ruficeps*)	2400–2800	Temperate mixed coniferous and decoduous forest	4	-	-	17.9–19.27 × 15.64–16.49	-	-	-	-	5–6	[19,27]
Indian Blue Robin (*Larvivora brunnea*)	1600–3300	Broadleaf evergreen forest, bamboo forest	4	Bryophyte, dead leaves, and wool	2.11–2.36	19.7–20.4 × 14.5–15.2	Plain pale blue	No speckles	≥13	12	5–6	[12]; this study
Siberian Blue Robin (*Larvivora cyane*)	1100–1500	Damp broadleaf evergreen and semi-evc rgreen bottomland forest with dense undergrowth, thickets	5–6	Bryophyte, dead leaves, and animal hair	1.8–2.2	15.0 × 19.5	Sky-blue to bright blue or greenish–blue	No speckles	12–13	13	5–6	[5,6,8]

There was a “bird blind” 200 m away from the nest, and the parent birds frequently brought worms to feed their nestlings. In the past 10 years, bird blinds, as emerging forms of ecotourism, have grown rapidly in China [37]. At present, they have been promoted from the Gaoligong mountains in Yunnan Province to other provinces and regions, including Sichuan, Guizhou, and Guangxi. In our case, the adult robin fetched worms many times for its nestlings. Extra feeding in the bird blind increases the food intake of the breeding birds, but subsequently facilitates the reproduction of birds nearby, which may attract birds to build nests nearby [38]. In fact, one nest of the Brown Parrotbill (*Cholornis unicolor*) and another nest of the White-Tailed Robin (*Myiomela leucura*) were also found nearby. However, this may impose negative effects on birds as well, including changes in their physical condition, breeding success, survival, community structure, and migratory behavior, as highlighted by previous studies [39,40,41,42]. In nature, birds may face increased predation risks at feeding sites [43,44]. Furthermore, feeding practices may also increase the likelihood of exposure to and the transmission of pathogens, such as prolonged high-density congregation, increased opportunities for inter-species mixing in habitats where such interactions are rare, and poor hygiene standards [45,46]. Finally, if the nutritional value of the food provided to birds is low or if it contains toxins that affect the host’s condition or immunity, this may also pose risks [46,47]. The long-term effects of bird blinds, including but not limited to avian reproduction, are still uncertain. It is imperative to maintain continuous monitoring and assessing the impacts of these man-made blinds on wild birds.

## 5. Conclusions

Drawing on life history theory, we have engaged in discourse concerning the solitary incubation and brooding behavior of the female Indian Blue Robin. Our study demonstrates how a small single-parent-care bird adjusts its incubation behavior, as well as the coordinated breeding of parents, to adapt to the cold and unpredictable external environmental changes in high-altitude regions. Despite only a single nest of the Indian Blue Robin being found, we present the first detailed account of the breeding biology of the Indian Blue Robin. Our findings revealed that the breeding season for this species commences in late May and concludes by the end of June. The Indian Blue Robin constructs its nest in ground cavities, and the clutch size is four. The eggs are oval, plain pale blue, and spotless. The incubation period is at least 13 days and the nestling period is 13 days. Both male and female parents are actively involved in the breeding process. Additionally, comparing our results with those of other bird species within the *Larvivora* genus, we find some discrepancies with the predictions of life history theory, which may be attributed to the limitation of our small sample size. The food supply in the bird blind has had a positive impact on the breeding of the Indian Blue Robin, providing extra food and reducing its reproductive pressure.

## Figures and Tables

**Figure 1 animals-14-00039-f001:**
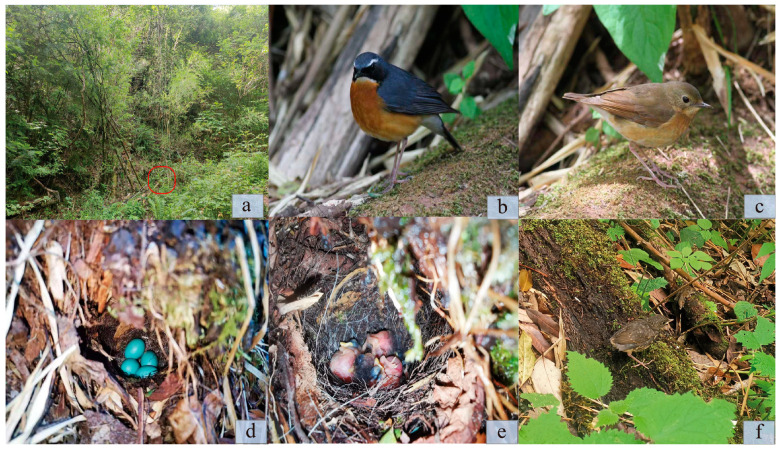
Photographic documentation of nesting habitat, breeding parents, eggs, nestlings, and fledglings of the Indian Blue Robin at the Laojun mountains, northwestern Yunnan, China. (**a**) Nest (circled) on the ground in hardwood and bamboo forests, (**b**) adult male, (**c**) adult female, (**d**) eggs, (**e**) one-day-old nestlings, (**f**) one-day-old fledgling.

**Figure 2 animals-14-00039-f002:**
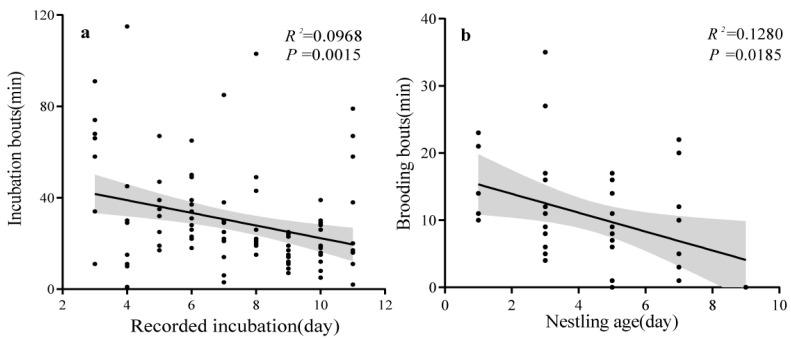
The incubation and brooding investment of female Indian Blue Robin. (**a**) Incubation bouts. (**b**) Brooding bouts.

## Data Availability

The data presented in this study are available upon request from the corresponding author. The data are not publicly available due to the rare number of the Indian Blue Robin thrushes; they need better protection.

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
