# Peer review of "First Account of the Breeding Biology of Indian Blue Robin (Larvivora brunnea) in Southwest China"

_animals, 2023, doi:10.3390/ani14010039_

Round 1

Reviewer 1 Report

Comments and Suggestions for Authors

Dear Authors,

I read with interest your manuscript entitled „First account of the breeding biology of Indian Blue Robin (Larvivora brunnea) in southwest China”. 

The manuscipt reports on the nesting of a bird species for which little information was available. Moreover, it comes from a geographical region where many similarly understudied bird species live. This is the strength of the article, which is not diminished by the fact that only one nesting was described. Given the declining populations of many songbird species in Asia, including those with ornamental plumage such as the species under study, any basic research is important for the knowledge and conservation of these species.

The article is well written and contains articles that could be considered so far. The second table is particularly useful, but does not include sample sizes, which would be important. Of course, if this is not stated in the source work, there is nothing to be done, but if it is, it should certainly be included. Unfortunately, the table is scattered, making it difficult to read, but this is just an editing error. Otherwise, I'm glad about the comparison with closely related species, because the authors have compiled all the information available.

I have pointed out a few small improvements below and have only one important observation. The extremely low sample size should be better emphasised at all points of the evaluation of the results, as you did in L247. Without this, comparisons cannot be made.

Minor comments:

L58: „Birds of the World” Please add the appropriate citation instead of the title of the book.

L73: „4,513 m” This data was mentioned later, it should be deleted from here.

L135: „Similarly, we used a similar model”: The word „similar” is redunant.

L205: „of this Robin” of this species

Comments on the Quality of English Language

Minor editing of English language required, for example the use of abbreviations (min instead of minute at first mention in the text).

Author Response

Dear Reviewer,
Thank you for taking the time to review my article. I have incorporated the changes you suggested and have uploaded the revised version in Word. Please let me know if you have any further comments or suggestions.
Best regards. Jun Nie

Reviewer 2 Report

Comments and Suggestions for Authors

In this study, the authors describe for the first time the breeding ecology of the XXX using data from a single nest. The information provided here is valuable as there is a notable gap of knowledge with regards to tropical bird species and thus I would like to encourage its publication. The work is well written and was conducted with care. I think however that the work has several limitations outlined below.

Disturbance and data quality: I’m afraid the authors exerted a negative influence on the nest. This is evident by the very short nesting time period reported. The figure of 13 days of nesting stage for a bird of this size is extremely short. The image provided in figure 1f indicates that this is bird is far from being ready to fledge, which suggests an early fledging event caused by human disturbance. Unfortunately, this compromises data quality and the least that can be done is acknowledging this limitation.

Limited comparison with related species: The discussion is quite well written but the authors barely discussed their data in relation to similar species. The only aspect that is addressed is the trade-off between clutch and egg size (and the text is a bit confusing). I would have expected a more careful discussion of the to what extent this species is similar to or differs from related species.

Minor comments:

Lines 49-50: describe sexual dimorphism as this is relevant to the study.

Line 58: “rough breeding season”? Do the authors mean “only the timing of breeding, (...) have been roughly described”? I would delete “roughly”.

Line 65: I would state “of any incubating bird” as the authors have not disclosed yet who is incubating. Also, please describe your hypothesis.

Lines 77-78: the wording employed here is weird, I would simply state that the site is an important biodiversity hotspot globally.

Line 90: should read “had started”

Lines 91-109: this entire section needs work. The authors need to explain better what the mean by “Bird Blind”, which sounds very weird. One needs to go to the discussion to understand what the authors were talking about. Please also provide more details on the timing of recordings and observations such as bout duration, time of the day, number of observers, etc. I would also like to note that daily visits including constant measurements of eggs and nestlings plus recording devices so close to the nest could be considered quite intense disturbance.

Lines 110-121: how many people was taking measurements in the field?

Line 120: should read “was absent”

Lines 122-124: I think this makes little sense. Extracting and transporting the nest would likely result in alterations to its shape. Why not taking these measurements in the field?

Lines 126-132: this is very difficult to judge based on the presented data. The wording employed is misleading as it is highly likely that the researchers were only able to obtain a snapshot of parental behavior.

Lines 133-140: when there is such an intense disturbance, it is common to leave a time for the parents to resume their normal behavior – this can be a certain period of time or e.g. waiting until one of the parents makes two visits. This would guarantee that your recordings are standardized to minimize the effect of human disturbance.

Figure 1:  I’m not an expert on tropical birds but what is shown in Figure 1f clearly is NOT a fledgling – that bird likely has reduced thermoregulatory and foraging capacities (and no tail feathers!!). I think the authors elicited an early fledging event due to the disturbance caused in the nest. This idea is supported by what is described in lines 183-186 and 192-194.

Line 173: should read “On average”

Line 188: Delete the first sentence – all nestlings grow “quickly”

Line 229: should read “by comparing”

Lines 236-237: this is confusing. Larger clutches or larger eggs?

Comments on the Quality of English Language

Grammar: while the text reads well, I noticed that present and past tense are used inconsistently throughout to report the results (which must be in past tense, e.g., in line 25 “is located”) and to describe general patterns (which must be in present tense, e.g., in line 38 “groups were understudied”), please revise. Sometimes words acting as adjectives should also be in past tense such as in line 42 “detail documentation” which should read “detailed”. This needs to be revised across the whole manuscript.

Author Response

尊敬的审稿人,
感谢您抽出宝贵时间审阅我的文章。我已经合并了您建议的更改,并已将修订后的版本上传到 Word 中。如果您有任何进一步的意见或建议,请告诉我。
此致敬意。聂军

Reviewer 3 Report

Comments and Suggestions for Authors

Due to cryptic behavior in the forest understory, and narrow distribution range, breeding data regarding Indian Blue Robin is notably sparse. This study provides the first, detailed information about the breeding biology of this cryptic species.

Line 93: we found a "Bird Blind" near the breeding nest, and the presence of the Bird Blind caused the Indian Blue Robin parents to frequently bring worms to feed their nestlings.

Please provide an explanation/description about "Bird Blind" here, rather than in Discussion.

Author Response

(The authors gave the same response as above.)

Reviewer 4 Report

Comments and Suggestions for Authors

I found your paper to be an informative note on a little known species. Your methods seem sound and your results and discussion are clear. My only concern was your use of a t-test to compare behavior of the parents. Since you only have a single nest, a t-test is not appropriate, and repeated measurements of the same two individuals is pseudoreplication. The remainder of my comments are grammatical, and everything is written on the manuscript. I wish you well in your efforts to publish this interesting piece.

Comments on the Quality of English Language

The English is pretty good, but there were some places, mostly the abstract and introduction, where I noted some corrections.

Author Response

(The authors gave the same response as above.)
